# Effect of High-Pressure Torsion on the Microstructure and Magnetic Properties of Nanocrystalline CoCrFeNiGa_x_ (x = 0.5, 1.0) High Entropy Alloys

**DOI:** 10.3390/ma15207214

**Published:** 2022-10-16

**Authors:** Natalia Shkodich, Franziska Staab, Marina Spasova, Kirill V. Kuskov, Karsten Durst, Michael Farle

**Affiliations:** 1Faculty of Physics and Center of Nanointegration (CENIDE), University of Duisburg-Essen, 47057 Duisburg, Germany; 2Physical Metallurgy, Materials Science Department, Technical University of Darmstadt, 64287 Darmstadt, Germany; 3Center of Functional Nano-Ceramics, National University of Science and Technology MISIS, 119049 Moscow, Russia

**Keywords:** high entropy alloy, high energy ball milling, spark plasma sintering, high pressure torsion, coercivity, Vickers hardness

## Abstract

In our search for an optimum soft magnet with excellent mechanical properties which can be used in applications centered around “electro mobility”, nanocrystalline CoCrFeNiGa_x_ (x = 0.5, 1.0) bulk high entropy alloys (HEA) were successfully produced by spark plasma sintering (SPS) at 1073 K of HEA powders produced by high energy ball milling (HEBM). SPS of non-equiatomic CoCrFeNiGa_0.5_ particles results in the formation of a single-phase *fcc* bulk HEA, while for the equiatomic CoCrFeNiGa composition a mixture of *bcc* and *fcc* phases was found. For both compositions SEM/EDX analysis showed a predominant uniform distribution of the elements with only a small number of Cr-rich precipitates. High pressure torsion (HPT) of the bulk samples led to an increased homogeneity and a grain refinement: i.e., the crystallite size of the single *fcc* phase of CoCrFeNiGa_0.5_ decreased by a factor of 3; the crystallite size of the *bcc* and *fcc* phases of CoCrFeNiGa—by a factor of 4 and 10, respectively. The lattice strains substantially increased by nearly the same extent. After HPT the saturation magnetization (*M*_s_) of the *fcc* phase of CoCrFeNiGa_0.5_ and its Curie temperature increased by 17% (up to 35 Am^2^/kg) and 31.5% (from 95 K to 125 K), respectively, whereas the coercivity decreased by a factor of 6. The overall *M*_s_ of the equiatomic CoCrFeNiGa decreased by 34% and 55% at 10 K and 300 K, respectively. At the same time the coercivity of CoCrFeNiGa increased by 50%. The HPT treatment of SPS-consolidated HEAs increased the Vickers hardness (*H*_v_) by a factor of two (up to 5.632 ± 0.188) only for the non-equiatomic CoCrFeNiGa_0.5_, while for the equiatomic composition, the *H*_v_ remained unchanged (6.343–6.425 GPa).

## 1. Introduction

In general, metallic materials currently available for most applications are based on one or two principal elements: Cu-based alloys, Ni-based superalloys, Fe-based steels, etc. [1,2]. Developing new alloys has mostly been focused on microstructure/surface modification and adding alloying elements to improve specific properties.

In 2004 Yeh et al. [3] showed that multi principal component alloys containing 5 or more principal elements in equiatomic or nearly equiatomic amounts (ranging between 5 and 35 at. %) could be processed to form simple solid solutions under appropriate conditions, called high entropy alloys (HEAs). Their high configurational entropy of mixing (especially at higher temperatures) improves the stability of chemically disordered phases with simple *bcc*, *fcc,* or *hcp* structures [3,4,5]. HEAs usually exhibit an excellent mechanical strength [6], ductility [7], and good thermal stability [5,8,9].

Recently, it has been demonstrated that certain HEAs containing elements with large magnetic moments possess very good magnetic properties [10,11,12,13,14] for the use as soft magnetic materials (SMM) with high saturation magnetization, high electrical resistivity, and malleability. Moreover, hard magnetic properties have been found in FeCoNiAlCu_x_Ti_x_-based HEAs [15] with *H*_c_ = 85.8 kA/m and a maximum energy product (BH_max_) of 16.39 kJ/m^3^, slightly better than the one of isotropically cast Alnico magnets. Furthermore, HEAs have also shown excellent perspectives for the usage as magnetocaloric materials [12].

The configurational space of alloying five or more different elements offers a huge playground to tune magnetic properties. For example, the addition of Cr to an equiatomic FeNiCo alloy which exhibits high saturation magnetization (*M*_s_) (151 Am^2^/kg) and low coercivity *H*_c_ = 120.96 A/m [16] significantly reduces the magnetization due to antiparallel coupling of the Cr spins to the Co/Fe/Ni spins and drives the CoCrFeNi alloy toward paramagnetism at room temperature [17].

Further addition of Al, Pd, or Ga promotes ferromagnetism and increases *M*_s_ from 0.5 to 13 Am^2^/kg, 34 Am^2^/kg and 38 Am^2^/kg and the Curie temperature to 277 K, 503 K and 703 K, respectively [10,17]. The enhancement was assumed to be due to the partial phase transition to a *bcc* phase.

In terms of processing the materials, melting, and casting routes have been most widely used, but the reproducibility and control of the sublimation of low-boiling elements turned out to be rather difficult, thus making compositional control difficult. At this, the grain size usually reached a value of several hundred micrometers. It was shown that different processing routes for the same material may result in the formation of distinct microstructures (e.g., grain size) with different properties. On the other hand, the high energy ball milling in planetary ball mills can yield stable microstructures and nanocrystalline HEAs with better homogeneity [8,18]. Recently, we reported on the HEBM fabrication of microsized fcc phase CoCrFeNiGa_x_ (x = 0.5, 1.0) particles with good structural and compositional homogeneity [19]. Using nanocrystalline milled HEA powders as a starting material prevents the grain growth in spark plasma sintering [19,20]. Moreover, shorter SPS times are required for consolidation of bulk materials relative to the conventional sintering procedure since the time window for grain growth is markedly shorter [21].

Another complementary, but top-down method which influences the microstructure and therefore the magnetic properties is the High-Pressure Torsion (HPT) process. It yields not only a grain refinement down to the nanoscale, but also can initiate various phase transformations, such as the formation or decomposition of supersaturated solid solutions, dissolution of phases, formation of nanocrystals during decomposition of amorphous phases or amorphization of crystalline phases [22,23,24,25]. Consequently, HPT can be used to tailor the magnetic properties by for example locally changing the alloy composition, yielding a pronounced grain refinement and strained grain interfaces as well as an increase in dislocation density [26,27]. The grain size can be reduced until a certain value, the saturation grain size. The typical saturation grain size for alloys and HEAs reported in the literature is in the range of 50–100 nm [28,29,30]. Recently, the effect of the HPT process on the saturation grain size and mechanical properties has been investigated for the Cantor alloy system [29,30]. In Reference [30] a single-phase nanocrystalline (grain size of 50 nm) CoCrFeNiMn alloy with few chromium oxide precipitates (of 7–10 nm) and high hardness of 6700 MPa has been produced by HPT.

In our search for an optimum soft magnet with the excellent mechanical properties we study the structure and magnetic properties of magnetic nanograined CoCrFeNiGa_x_ (x = 0.5, 1.0) bulk alloys starting from the HEBM powders [19], which are consolidated by spark plasma sintering, and subsequently processed by HPT.

## 2. Materials and Methods

CoCrFeNiGa_x_ (x = 0.5, 1.0) HEA powders were prepared by high energy ball milling (HEBM) of commercial powders of Co (99.97% pure, particle size ~45 µm), Cr (99.7% pure, particle size 10–30 µm), Fe (99.96% pure, particle size < 150 µm), Ni (99.5% pure, particle size 45–60 µm), and Ga (99.99% pure, ingot) taken in aliquot amounts. For the equiatomic CoCrFeNiGa composition, 20 at. % of each element was taken, whereas for the CoCrFeNiGa_0.5_ alloy: Co 22.2, Cr 22.2, Fe 22.2, Ni 22.2, and Ga 11.2 at. %.

HEBM was performed in a water-cooled planetary ball mill Activator-2S using stainless steel cylindrical jars and steel balls (7 mm in diameter). In all cases, the ball/powder weight ratio was 20:1. The vial was evacuated and then filled with Ar gas up to 4 bars. The HEBM was run at a rotating speed of the sun wheel and the grinding drums at 900 and 1800 rpm, respectively. The milling time (t) in Ar reached 180 min. An additional t = 10 min (in C_3_H_7_OH) of milling after HEBM in Ar was applied.

The non-milled and milled CoCrFeNiGa_x_ (x = 0.5, 1.0) powders were SPS-consolidated in vacuum in a Labox 650 facility (Sinter Land, Japan). The powder mixture was placed into a cylindrical graphite die (inner diameter 12.7 mm) and uniaxially compressed at 10–50 MPa. The sample was heated at a rate of 100 K/min up to 1073 K by passing rectangular pulses of electric current through the sample. The dwell time at sintering temperature was 10 min. SPS-produced disks were 2–3 mm thick and 12.7 mm in diameter.

A set of SPS-consolidated samples (disks of 10 mm in diameter) were severely deformed by 10 rotations (6 rotations for the HEBM sample CoCrFeNiGa_0.5_, Table 1) in high pressure torsion at room temperature (RT). The process was conducted at a rate of 1 rpm and an applied contact pressure of 5.1 GPa, using water cooling to keep the temperature close to RT (30–40 °C). A schematic diagram of the processes is presented in Figure 1.

The preparation conditions for both compositions are summarized in Table 1.

Initial and milled powders, SPS-consolidated and HPT-deformed samples were characterized by X-ray diffraction (XRD) using Fe-Kα radiation to distinguish structures of Co, Fe, and Ni (DRON-3M diffractometer, Fe-Kα radiation with λ = 0.19374 nm, 2θ = 40–120°). Crystalline phases were identified using Crystallographica Search-Match 2.1 software and ICDD PDF2 database. To determine crystal cell parameters, crystallite size and strains we used the PDWin 6 software (NPP Bourevestnik, Saint-Petersburg, Russia, 2010) [31]. For the calculations of the crystalline phase content, the Rietveld method was used.

SPS-consolidated and HPT-processed samples were cross sectioned along the diameter and polished to analyze their microstructure and chemical composition by scanning electron microscopy (SEM, with operating voltage 15 kV) with operating voltage 15 kV along with energy dispersive spectroscopy (EDX) (Zeiss Ultra+ microscope + Oxford Inca spectrometer, Aztec 2.1 software). Magnetic properties of the samples were determined using a Quantum Design DynaCool Physical Property Measurement system (PPMS) at temperatures in the range of 5–390 K under external magnetic fields up to 9 Tesla.

The microhardness of SPS-consolidated/HPT-deformed CoCrFeNiGa_x_ (x = 0.5, 1.0) samples was measured using Vickers hardness tests with an Emco-Test DuraScan 70 (Austria) under an applied load of 4.9 N (HV_0.5_).

## 3. Results and Discussion

### 3.1. Structural Characterization of SPS-Consolidated and HPT-Processed CoCrFeNiGax (x = 0.5, 1.0) HEAs

We focused our study on tuning the structure and magnetic properties by HPT for bulk CoCrFeNiGa_x_ (x = 0.5, 1.0) HEAs produced by SPS from HEBM powders (190 min at 900/1800 rpm), as bulk HEAs from elemental powders Co, Cr, Fe, Ni, and Ga ingots by SPS could not be fabricated (for details, see Appendix A).

For SPS consolidation and subsequent HPT treatment the CoCrFeNiGa_0.5_ and CoCrFeNiGa HEA powders with a *fcc* lattice parameters (calculated as a = 3.610 ± 0.005 Å and a = 3.632 ± 0.007 Å, respectively), and a uniform distribution of principal elements obtained by HEBM were used [19]. The details of their structural and magnetic behavior after HEBM are described in [19].

Figure 2 shows XRD data of SPS-consolidated (dash red) and HPT-deformed (blue) CoCrFeNiGa_0.5_ HEAs produced from HEA powders (black).

SPS-consolidation of HEBM-processed HEA powder (Figure 2, black) at 1073 K leads to an increase in crystallinity of the *fcc* phase (Figure 2, dash red). The XRD (111), (200), and (220) peaks become narrower after sintering due to an increase in the crystallite size and decrease in lattice strains caused by HEBM. Other additional phases were not observed. The subsequent HPT treatment of the SPS-consolidated HEA (Figure 2, dash red) led to a decrease in intensity of XRD (111), (200), and (220) peaks (Figure 2, blue) due a grain refinement and a substantial increase in lattice strains of the CoCrFeNiGa_0.5_
*fcc* single-phase (Table 2). The phase transformation has not been occurred in HPT-deformed HEA (Figure 2, blue).

The SEM/EDX results for both the SPS-consolidated and the subsequently HPT-deformed CoCrFeNiGa_0.5_ HEAs show that the elements of Co, Cr, Fe, Ni, and Ga are primarily uniformly distributed on a micro-scale (Figure 3a,b). Black Cr-rich precipitates were also detected by EDX. Some traces of laminar flow structures and increased alloy homogeneity were observed caused by HPT (Figure 3b).

SPS consolidation of the equiatomic CoCrFeNiGa HEA powder (Figure 4, black) at 1073 K led to partial decomposition of the *fcc* structure. XRD data showed that SPS-consolidated material (Figure 4, dash red) contained a mixture of *bcc* and *fcc* phases, those crystallite sizes decreased, and lattice strains increased after HPT treatment (Figure 4, blue, Table 2). It is important to note that there is no evidence for appearance of a new phase and the occurrence of the phase transformation during HPT processing.

The SEM/EDX results of SPS-consolidated and HPT-deformed CoCrFeNiGa HEAs (Figure 5) show a uniform equiatomic distribution of the principal elements. Almost spherical grains (Figure 5a) transform into elongated grains (Figure 5b). The presence of few Cr-rich precipitates for both HEAs was also detected; however, it is evident that the interfaces of these precipitates are less sharp after the HPT process.

Our previous work [19] showed that the crystallite sizes of single-phase *fcc* HEBM-processed HEA CoCrFeNiGa_x_ (x = 0.5, 1.0) powders were around 9 nm. The microstrains were 0.12% and 0.36% for CoCrFeNiGa and CoCrFeNiGa_0.5_, respectively.

The crystallite size and strain in bulk CoCrFeNiGa_x_ (x = 0.5, 1.0) was derived from the line width analysis of XRD peaks. The pseudo-Voigt function was used for fitting of the XRD peak profile, and a Si standard was used to correct for instrumental broadening. Crystallite size, lattice strain, and lattice parameters for *bcc* and *fcc* phases were calculated for SPS-consolidated and HPT-deformed CoCrFeNiGa_x_ (x = 0.5, 1.0) HEAs using the method of second central moments [31]. The results of the calculations are summarized in Table 2.

The crystallite size of the single-phase *fcc* CoCrFeNiGa_0.5_ HEA powder [19] increases by a factor of 6 after SPS-consolidation and decreases by a factor of 3 after HPT treatment which is typical for such processes. The lattice strains show opposite behavior—reduced after SPS-consolidation at 1073 K and increased again during HPT deformation (Table 2).

For the equiatomic SPS-consolidated CoCrFeNiGa HEA the crystallite size of both *bcc* and *fcc* phases decreases by a factor of 4 and 10 after HPT, respectively. Microstrains of *fcc* and *bcc* phases in HPT-deformed CoCrFeNiGa HEA increased to nearly the same extent (see Table 2). It has to be mentioned that the measured strain of 1.73% after HPT is probably not the true strain but the maximum strain due to defects and additional broadening mechanisms. A slight increase in crystal lattice for both *fcc* and *bcc* structures in the bulk HEAs (Table 2) after HPT can be attributed to the partial dissolution of the precipitates previously observed in SPS-consolidated HEAs (Figure 3 and Figure 5).

The values of volume-to-volume ratio of *bcc* (~84%, R_wp_ = 6.87%) to *fcc* (~16%, R_wp_ = 6.87%) solid solutions in SPS-consolidated CoCrFeNiGa alloy slightly changed after the HPT deformation: the amount of a *bcc* phase decreased (~75%, R_wp_ = 7.76%), while the amount of *fcc* one increased (~25%, R_wp_ = 7.76). This phase transformation can be linked to the deformation during the HPT process, and was recently also reported for HEAs in the literature [32,33,34].

SEM-BSE micrographs (Figure 6) at higher magnification (×40,000) show that SPS-consolidated CoCrFeNiGa_0.5_ HEA (Figure 6a) undergoes nanostructuring (drastic decrease in grain size) and an increased homogeneity upon HPT without any significant severe grain flow (Figure 6b). For equiatomic CoCrFeNiGa bulk (Figure 6c), a lamellar microstructure with a pronounced two-phase separation was observed after HPT. The SEM-BSE results agree well with the XRD data (Figure 3 and Figure 4).

In Figure 6 the Ga-rich areas (Z = 31) look brighter than the other elements: Cr (Z = 24), Co (Z = 27), Fe (Z = 26), and Ni (Z = 28). We assume that the darker areas on Figure 6d correspond to the Ga-depleted phase with the *fcc* structure (Figure 4) and the brighter regions to the Ga-rich *bcc* phase.

### 3.2. Magnetic Properties

#### 3.2.1. CoCrFeNiGa_0.5_ HEAs

HPT deformation of the SPS-consolidated CoCrFeNiGa_0.5_ HEA leads to an increase of the saturation magnetization and the Curie temperature (Figure 7). A very broad transition from a ferro- to a paramagnetic state is observed. The magnetization of both samples (Figure 7a) decreases already at temperatures much lower than the approximate Curie temperature *T*_c_. This behavior of M(T) differs from classical ferromagnets where a significant decrease in the magnetization starts at T > 2/3 *T*_c_ followed by a rapid loss of spontaneous magnetization at the Curie temperature. Smoothing of the transition across such a wide temperature range indicates magnetic inhomogeneities which are possibly due to a broad distribution of local magnitudes and signs of exchange interactions as a consequence of disorder on the atomic scale.

An apparent Curie temperature can be determined according to the Landau theory as a minimum of the first derivative dM/dT (Figure 7b). This point marks a crossover from an exchange-ordered to a field-ordered magnetic state [35]. Using this model, we find that HPT results in an increase of this apparent Curie temperature from 95 K to 125 K.

Also, a “paramagnetic ordering temperature” $\theta_{C}$ can be estimated from the Curie–Weiss law as the temperature at which the paramagnetic inverse susceptibility 1/χ is zero.

Well above $\theta_{C}$ in the paramagnetic state, the temperature dependence of the susceptibility is described by the Curie–Weiss law: $\chi \left(T\right)=\frac{C}{T-\theta_{C}}$, where *C* is the Curie constant $C=n\frac{{µ}_{0}}{3k_{B}}{µ}_{eff}^{2}$. Here *n* is the volume density of atoms, *μ*_0_ is the permeability of free space, $k_{B}$ is Boltzmann’s constant and ${µ}_{eff}^{2}$ is the squared effective magnetic moment per atom. An effective magnetic moment per atom can be defined using the formula: $\mu_{eff}=\sqrt{\frac{nC\mu_{0}}{3k_{B}}}$. A fit according to the Curie–Weiss law (green dash lines in Figure 7c) yields the average moment per atom $1.52{µ}_{B}$ for both samples: before and after HPT treatment and a “paramagnetic” Curie temperature, $\theta_{C}$, of 125 K and 185 K, respectively. The atomic density $\sim 8.57\times 10^{28}{\;m}^{-3}$ was estimated based on the results of XRD studies. Note that the “paramagnetic” Curie temperature, $\theta_{C}$, is much higher than the “ferromagnetic” temperature we have determined earlier (Figure 7b). We assume that above “*T*_c_” the long-range ferromagnetic correlations observed as M(T) are stabilized by an external magnetic field.

Conventionally, the average magnetic moments per atom can be determined from the low temperature saturation magnetization. We find ~0.31${µ}_{B}$ and ~0.36${µ}_{B}$ for SPS consolidated alloys before and after HPT, respectively. The large discrepancy between the magnetic moments determined from the low temperature magnetization (0.31${µ}_{B}$) and the Curie–Weiss-law (1.52${µ}_{B}$) indicates the presence of antiferromagnetic exchange interactions which are thermally overcome in the paramagnetic regime. Thus, we may conclude that Cr is antiferromagnetically correlated to neighboring elements in the equiatomic CrFeCoNi alloy [36,37]. Note that $\theta_{C}$ reflecting the dominance of thermal energy over the exchange correlations is much higher than *T*_c_.

How to interpret the discrepancy between the magnetic moments determined from the saturation magnetization and in the paramagnetic regime? In the ferromagnetic state we determine the average magnetic moment given by the band structure, whereas in the paramagnetic mode we assume the so-called local magnetic moment, which may in turn deviate from the present atomic moment due to ferromagnetic interatomic correlations still existing above the Curie temperature [38].

In addition, antiferromagnetic exchange interatomic interactions are expected in CoCrFeNiGa alloy, predominantly at the Cr sites, similar to those observed in the equiatomic CrFeCoNi alloy [36,37].

The deviation of 1/χ(T) above 250 K from the paramagnetic behavior for the as-prepared CoCrFeNiGa_0.5_ HEA can be attributed to the presence of superparamagnetic precipitates with a blocking temperature of 250 K which give rise to an effective paramagnetic susceptibility. HPT of the CoCrFeNiGa_0.5_ results in the dissolution of these precipitates and the formation of a more homogeneous alloy as was observed in SEM/EDX studies (Figure 3b). The field dependence of the magnetization of SPS-consolidated and HPT-deformed CoCrFeNiGa_0.5_ HEAs (Figure 8b) confirms this interpretation: dissolution of the precipitates which act as pinning centers for domain walls results in magnetic softening of the alloy after HPT processing.

#### 3.2.2. CoCrFeNiGa High Entropy Alloy

Figure 9 shows temperature dependences of magnetization for SPS-consolidated and HPT-deformed equiatomic CoCrFeNiGa HEAs. In contrast to CoCrFeNiGa_0.5_ the equiatomic alloy shows a higher saturation magnetization and a *T*_c_ well above 390 K. The increase in M(T) below 40 K approximately indicates the presence of an additional ferromagnetic phase with an ordering temperature of ~22 K and a magnetization which is about 2 Am^2^/kg, i.e., ~4.5% of the other phase (blue curve).

HPT processing results in a significant decrease in the overall magnetization. The low-temperature ferromagnetic phase manifests itself more clearly, and its Curie temperature increases up to 40 K. A larger magnetic inhomogeneity of the high-Curie temperature phase is evidenced by the smoother transition to the paramagnetic state.

HPT processing of the equiatomic CoCrFeNiGa HEA results in a decrease in the saturation magnetization by 34% and 55% (see Table 3) at 10 K and 300 K, respectively (Figure 10). The decrease in *M*_s_ is due to the increased volume fraction of the Ga-depleted *fcc* phase, which has a lower magnetization (Figure 8a) compared to the *bcc* Ga-enriched phase [17].

Note that the coercivity *H*_c_ of both samples is higher at room temperature than at 10 K. Above the Curie temperature of the *fcc* phase, ferromagnetic exchange between *bcc* nanocrystalline ferromagnetic grains is weakened resulting in magnetic hardening [39]. *H*_c_ (300 K) for the HPT deformed alloy is larger by 50% in comparison to the SPS-consolidated one. This increase can be attributed to the introduction of defects which act as pinning centers for domain wall movement.

### 3.3. Mechanical Properties (Vickers Microhardness)

The Vickers hardness (*H*_v_) for the SPS-consolidated and HPT-deformed CoCrFeNiGa_x_ (x = 0.5, 1.0) samples from elemental powder blends showed the lowest *H*_v_ with significant variations for both compositions (for details, see Appendix A).

The dependence of the Vickers hardness (*H*_v_) before and after HPT for SPS-consolidated samples obtained from HEBM CoCrFeNiGa_x_ (x = 0.5, 1.0) powders [19] are presented in Figure 11.

The *H*_v_ of the SPS-consolidated CoCrFeNiGa and CoCrFeNiGa_0.5_ HEAs (HEBM processed) is 3 and 5 times higher than for bulk samples sintered from elemental powder blends (see Appendix A), respectively. It is worth noting, that the subsequent HPT treatment of SPS-consolidated HEAs increased *H*_v_ by a factor of two (up to 5.632 ± 0.188 GPa) only for the non-equiatomic CoCrFeNiGa_0.5_, while for the equiatomic composition, *H*_v_ remained unchanged. The high hardness for SPS-consolidated equiatomic CoCrFeNiGa HEAs may be attributed to the mixture of *fcc* and *bcc* phases, and Cr-rich precipitates which hinders the gliding of dislocations and lattice planes at grain boundaries. Subsequent refinement of the microstructure by applied shear strain (HPT) does not significantly change the microstructure of grains. Only a slight elongation of the grains was observed (Figure 11).

As expected, an insignificant gradient in hardness along the radius of the disk is visible as for SPS-consolidated as for HPT-deformed samples (Figure 11).

A slight decrease in *H*_v_ observed at the edges of the SPS-consolidated samples caused by the specific features of SPS processing. Due to the difference in electrical conductivity between the sintered material and electrically conducting graphite die, more electric current pulses pass through the die than through the sample. Thus, the sintering temperature at the edges is slightly higher, causing faster grain growth. Therefore, the *H*_v_ at the edges is slightly smaller than in the center for all SPS-consolidated HEAs.

For the HPT-deformed bulk samples, on the contrary, a slight increase in hardness (Figure 11, blue) was observed from the center to edge due the expected larger imposed strain by torsion at the edge of the disk than in the center, which is typical for materials processed by high-pressure torsion [30]. SEM/EDX results of the HPT-deformed CoCrFeNiGax (x = 0.5, 1.0) HEAs show that the microstructural changes in the edge and the center of the sample are negligible, and the chemical composition with the uniform distribution of the elements are preserved (for details, see Appendix A).

## 4. Conclusions

Nanocrystalline CoCrFeNiGa_x_ (x = 0.5, 1.0) bulk HEAs have been successfully produced by SPS of HEBM-produced HEA powders and subsequently deformed by HPT. With the aid of the HPT deformation, we can tune the magnetism, since we can modify the microstructure and the local composition of the HEA which has an influence on the magnetic properties.

(1)The use of HEBM produced CoCrFeNiGa_x_ (x = 0.5, 1.0) powders allowed the synthesis of a homogeneous bulk HEA, which was not possible starting with elemental powders of Co, Cr, Fe, Ni, and Ga ingots in a direct SPS process, and even with the subsequent HPT treatment. The CoCrFeNiGa_0.5_ forms a single-phase *fcc* alloy whereas the equiatomic one shows two phases: Ga-depleted *fcc* and Ga-enriched *bcc* phases(2)SPS at 1073 K of the CoCrFeNiGa_0.5_ powder increased the crystallinity of the *fcc* phase, while for the equiatomic CoCrFeNiGa powder a partial transformation of the *fcc* structure into a *bcc* one was observed.(3)HPT led to a grain refinement for both compositions: the crystallite size of the *fcc* phase for CoCrFeNiGa_0.5_ bulk HEAs decreased by a factor of 3; the crystallite size of *bcc* and *fcc* phases for CoCrFeNiGa bulk ones—by a factor of 4 and 10, respectively. The lattice strains substantially increased to nearly the same extent.(4)HTP processing of the single *fcc* phase CoCrFeNiGa_0.5_ HEA led to a 31% increase in the Curie temperature. The saturation magnetization increased by 17% (up to 35 Am^2^/kg) at 10 K. A decrease in the coercivity during the HPT can be attributed to the dissolution of precipitates.(5)HPT processing of the equiatomic CoCrFeNiGa resulted in an increase in the *fcc* fraction with a low Curie temperature and enhanced magnetic inhomogenity of the ferromagnetic bcc phase. The overall saturation magnetization of the sample decreases: by 34% at 10 K and 55% at 300 K. The HPT increased the coercivity *H*_c_ (300 K) by 50% for the CoCrFeNiGa bulk HEA.(6)*H*_v_ (up to 5.632 ± 0.188 GPa) doubled for the single *fcc* phase SPS-consolidated CoCrFeNiGa_0.5_ HEA after HPT, while equiatomic SPS-consolidated and HPT-deformed CoCrFeNiGa HEAs showed a maximum value of *H*_v_ in the range of 6.343–6.425 GPa.

## Figures and Tables

**Figure 1 materials-15-07214-f001:**
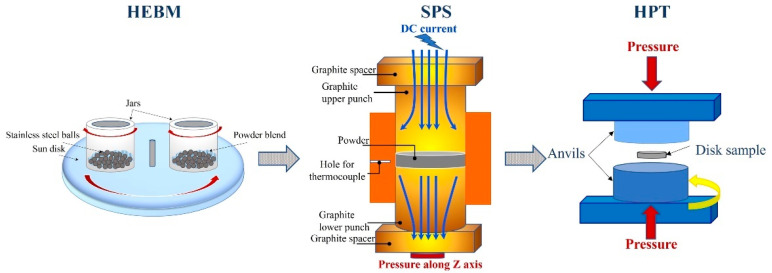
A schematic diagram of the combined use of HEBM, SPS and HPT processes for the synthesis of nanocrystalline bulk HEAs.

**Figure 2 materials-15-07214-f002:**
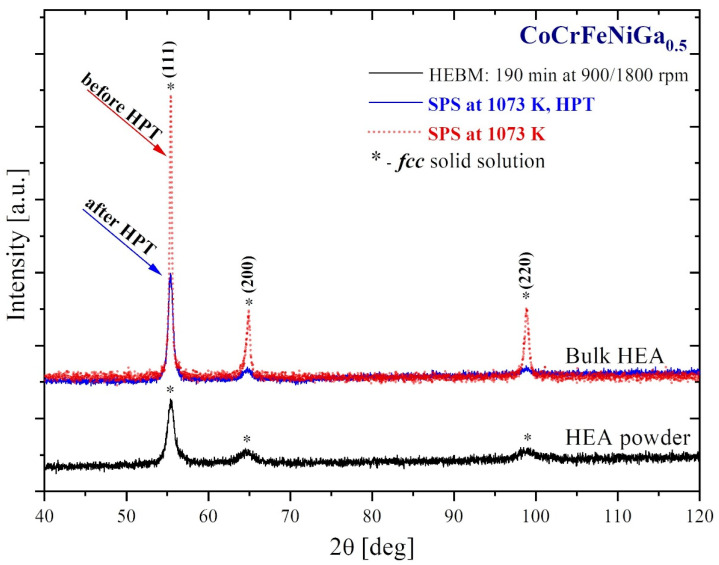
XRD patterns for CoCrFeNiGa_0.5_ HEBM-processed HEA powder (black), SPS-consolidated (dash red), and HPT-deformed (blue) samples.

**Figure 3 materials-15-07214-f003:**
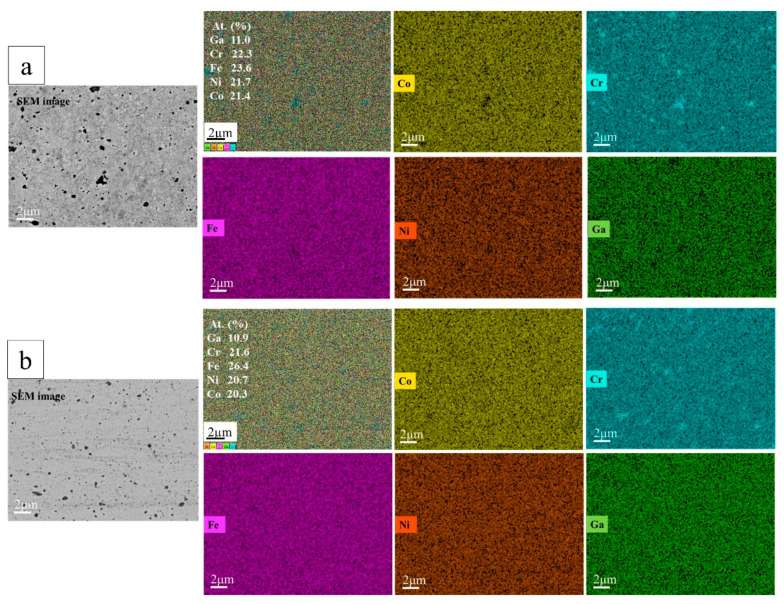
SEM images and EDX-mapping of a cross-section of bulk CoCrFeNiGa_0.5_ HEAs (**a**) sintered from HEBM powder blend by SPS at 1073 K (**b**) after subsequent HPT treatment.

**Figure 4 materials-15-07214-f004:**
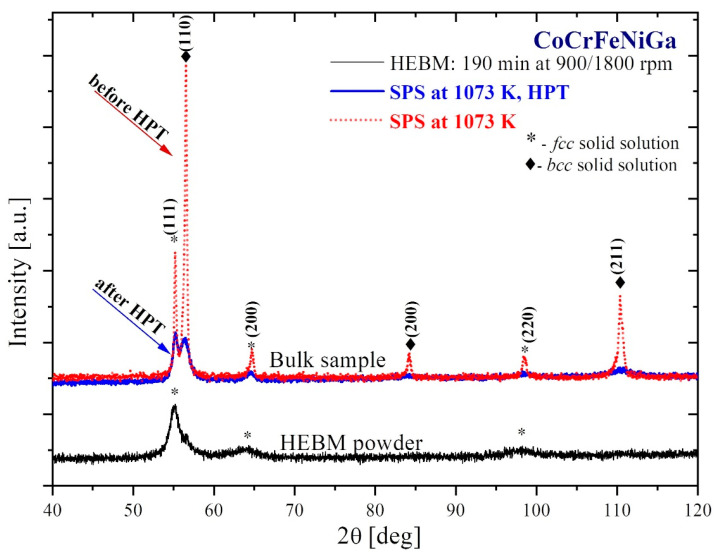
XRD patterns of the CoCrFeNiGa HEBM-processed powder (black), SPS-consolidated (dash red), and HPT-deformed (blue) samples.

**Figure 5 materials-15-07214-f005:**
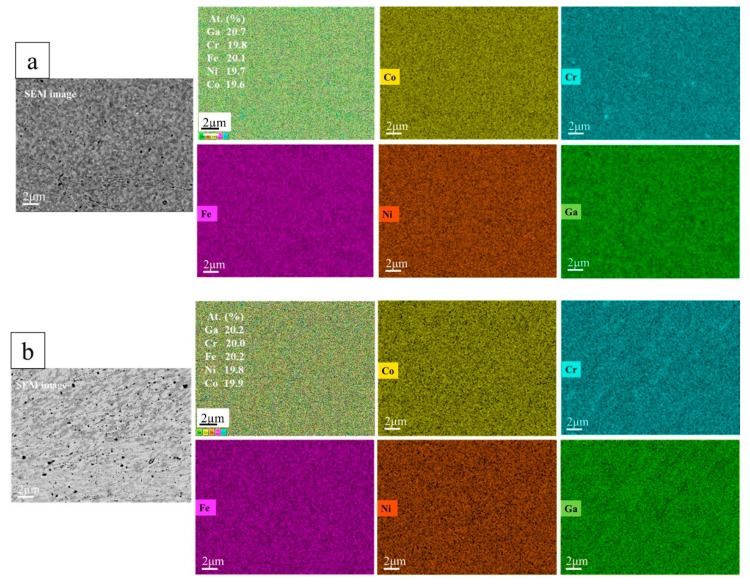
SEM images and EDX-mapping of the cross-section of bulk CoCrFeNiGa alloy (**a**) SPS-consolidated from HEA powder blend at 1073 K (**b**) after subsequent HPT treatment.

**Figure 6 materials-15-07214-f006:**
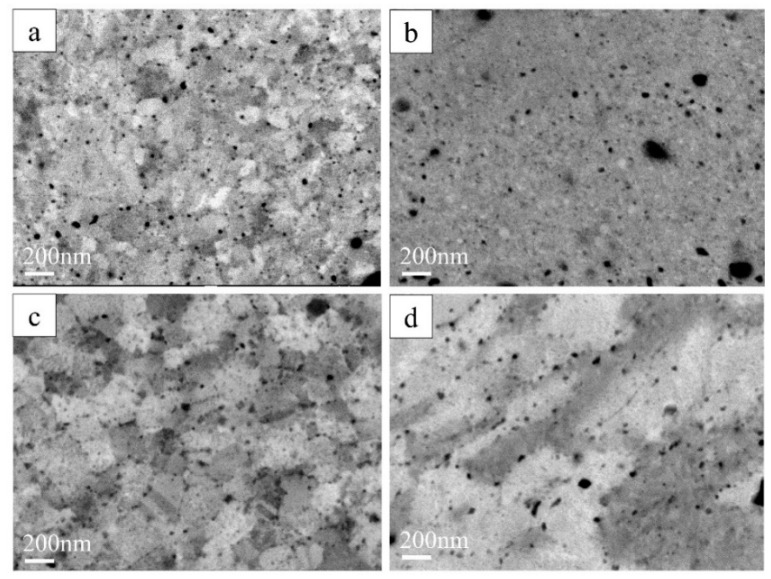
SEM-BSE images of (**a**,**b**)—bulk CoCrFeNiGa_0.5_ HEA before and after HPT, respectively; (**c**,**d**)—bulk CoCrFeNiGa HEA before and after HPT, respectively.

**Figure 7 materials-15-07214-f007:**
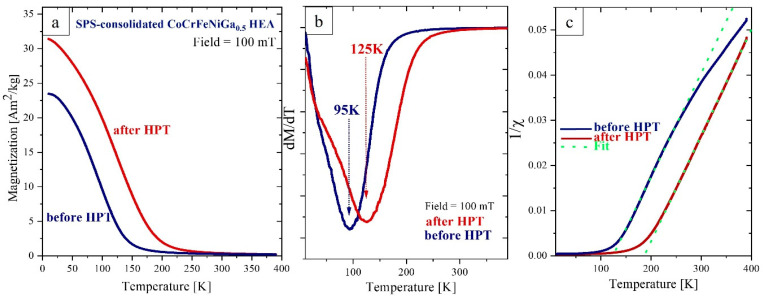
(**a**) Temperature dependent magnetization M(T) of SPS-consolidated CoCrFeNiGa_0.5_ recorded at 100 mT before (blue) and after (red) HPT treatment. (**b**) The slope of M(T) yielding an estimate for the Curie temperature at its minimum. (**c**) Inverse susceptibility versus temperature. Dash lines are fits according to the Curie–Weiss law. Note the change in slope for the blue curve above 260 K (for details see text).

**Figure 8 materials-15-07214-f008:**
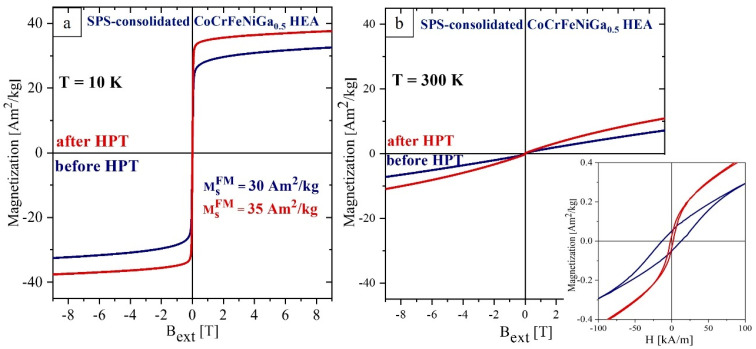
Field dependence of magnetization recorded at 10 K (**a**) and at 300 K (**b**) for SPS-consolidated CoCrFeNiGa_0.5_ HEA before (blue) and after (red) HPT. Insert M(H) at low fields.

**Figure 9 materials-15-07214-f009:**
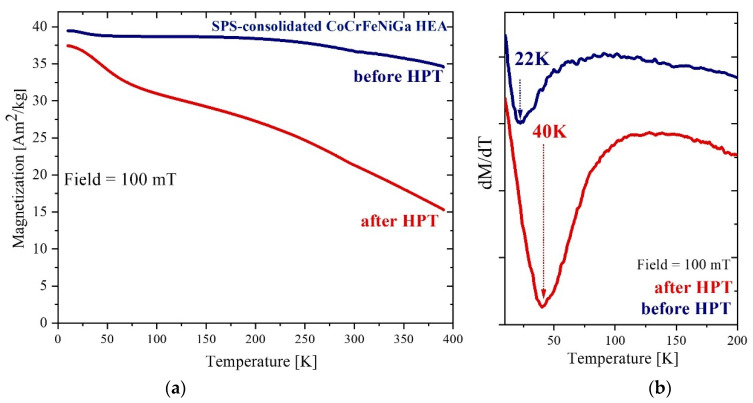
(**a**) Temperature dependence of magnetization measured at 100 mT for SPS-consolidated (blue) and after HPT treatment (red) CoCrFeNiGa HEA. (**b**) Temperature dependences of the slopes of M(T) (dM/dT). The Curie temperature is estimated at the minimum in plot (**b**).

**Figure 10 materials-15-07214-f010:**
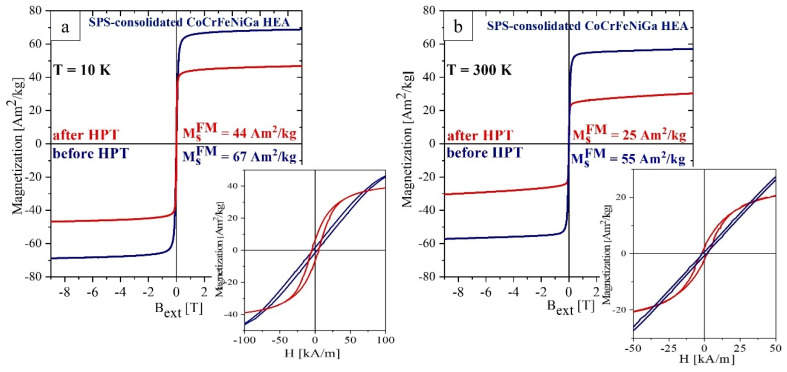
Field dependence of magnetization at 10 K (**a**) and 300 K (**b**) for SPS-consolidated (blue) and after HPT treatment (red) CoCrFeNiGa. Inserts show the low-field region of the M(H) curves.

**Figure 11 materials-15-07214-f011:**
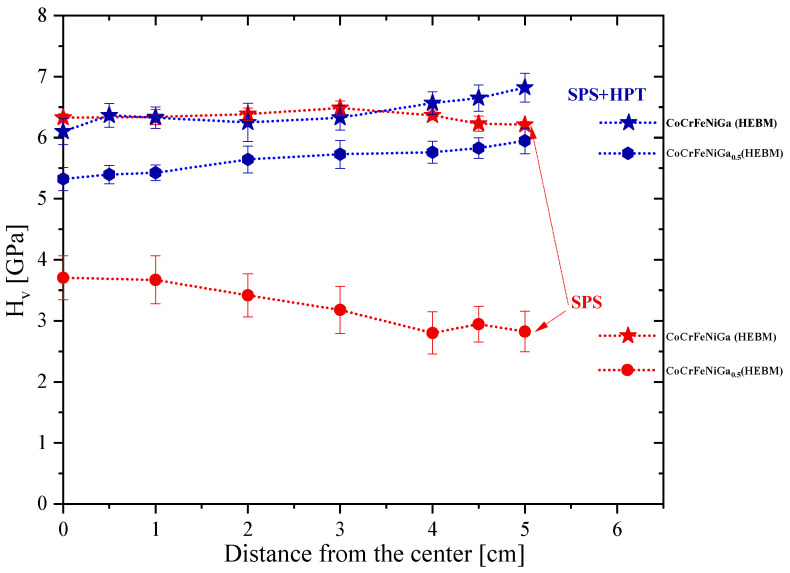
Dependence of the Vickers hardness on the radial distance from the center for SPS-consolidated (red), and subsequently HPT-deformed (dark blue) CoCrFeNiGa_x_ (x = 0.5, 1.0) samples from HEBM powder mixtures.

**Table 1 materials-15-07214-t001:** Preparation conditions for CoCrFeNiGa_x_ (x = 0.5, 1.0) alloys.

Composition	HEBM		SPS			HPT		
	*t*, min	Speed Ratio Sun Wheel/Jars, rpm	T, K	P, MPa	Dwell Time, min	Thickness, mm	Rotations	Mises Equivalent Strain ε
CoCrFeNiGa	No milling		1073	10	10	1.30	10	139
CoCrFeNiGa	180 + 10 *	900/1800	1073	50	10	0.85	10	213
CoCrFeNiGa_0.5_	No milling		1073	10	10	1.35	10	134
CoCrFeNiGa_0.5_	180 + 10 *	900/1800	1073	50	10	1.00	6	108

* 10 min in isopropanol (C_3_H_7_OH).

**Table 2 materials-15-07214-t002:** Crystallite size, lattice strain, and lattice parameters for bulk CoCrFeNiGa_x_ (x = 0.5, 1.0) HEAs before and after HPT.

		Before HPT	After HPT	Before HPT	After HPT
		CoCrFeNiGaHEBM: 180 + 10 min, 900 rpm SPS 1073 K, 50 MPa	CoCrFeNiGa HEBM: 180 + 10 min, 900 rpm SPS 1073 K, 50 MPa	CoCrFeNiGa_0.5_ HEBM: 180 + 10 min, 900 rpm SPS 1073 K, 10 MPa	CoCrFeNiGa_0.5_ HEBM: 180 + 10 min, 900 rpm SPS 1073 K, 10 MPa
Crystallite size, nm	*fcc*	30 ± 5	3 ± 0.5	54 ± 9	15 ± 2
	*bcc*	43 ± 7	10 ± 1.7	-	-
Strain, %	*fcc*	0.14 ± 0.02	1.73 ± 0.29	0.10 ± 0.01	0.31 ± 0.05
	*bcc*	0.08 ± 0.01	0.54 ± 0.09	-	-
*a*, Å	*fcc*	3.616 ± 0.001	3.622 ± 0.001	3.605 ± 0.001	3.611 ± 0.001
	*bcc*	2.887 ± 0.001	2.891 ± 0.003	-	-

**Table 3 materials-15-07214-t003:** Magnetic parameters—*M*_s_ and *H*_c_—for CoCrFeNiGa_x_ (x = 0.5, 1.0).

	CoCrFeNiGa_0.5_ HEA		CoCrFeNiGa HEA	
	SPS	HPT	SPS	HPT
***M*_s_ (10 K) [Am^2^/kg]**	30	35	67	47
***M*_s_ (300 K) [Am^2^/kg]**	0.1	0.1	55	25
***H*_c_ (10 K) [A/m]**	<50	<50	2000	2000
***H*_c_ (300 K) [A/m]**	13,200	2000	3200	4800

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
