# Peer review of "Effect of High-Pressure Torsion on the Microstructure and Magnetic Properties of Nanocrystalline CoCrFeNiGax (x = 0.5, 1.0) High Entropy Alloys"

_materials, 2022, doi:10.3390/ma15207214_

Round 1
Reviewer 1 Report
The article is devoted to the study of the properties of CoCrFeNiGax high-entropy alloys, as well as their characteristics, which were obtained by spark plasma sintering. In general, the presented study is quite interesting and promising not only from a fundamental point of view, but also from a further practical application. The article corresponds to the declared journal and can be accepted for publication in the future after the authors answer a number of questions that have arisen during its analysis.
1. The abstract needs to be improved, the authors should reflect in more detail the novelty and practical significance of the work.
2. The presented diffraction patterns require significant processing in presentation and subsequent interpretation.
3. The authors should give a more detailed explanation of the effect of deformation of the crystal lattice with a decrease in the size of crystallites, as well as their relationship with a change in the phase composition.
4. The authors should explain the differences in hardness values ​​depending on the measurement distance, the data obtained indicate the heterogeneity of the composition.
Author Response
We thank the Reviewer for the valuable and helpful comments and overall positive evaluation of the manuscript - with a recommendation for acceptance. We revised the manuscript according to the suggestions of the reviewers. Below, you will find our point-by-point response. The respective changes in the text are marked in yellow We hope that the revised version of the manuscript can be accepted for publication without further delay.
Please see the attachment.

Reviewer 2 Report
Natalia et al., have conducted a study entitled, “Effect of high pressure torsion on the microstructure and magnetic properties of nanocrystalline CoCrFeNiGax (x = 0.5, 1.0) 3 high entropy alloys”, where they reported on nanocrystalline CoCrFeNiGax (x = 0.5, 1.0) bulk HEAs which have been successfully produced by SPS of HEBM-produced HEA powders by high energy ball milling method and subsequently deformed by HPT with the aid of the HPT deformation. The author claims to tune the magnetism as they can modify the microstructure and the local composition of the HEA which has an influence on the magnetic properties. It is an excellent piece of research that has applications in the fields of magnetocaloric materials and so on. Because MDPI Materials is a hard-core journal in the field of physics, materials science, and semiconductors, I would urge that this research be published with the following criticisms addressed.
1- Line # 14, the abbreviation HEA is inconsistent with line # 42, please use HEA or HEAs.
2- Author has claimed the single phase fcc bulk HEA, how did they confirm the single of the materials?
3- Please state the reason in the reduction of the crystallite size reduction? Usually TEM is the most reliable technique to measure the crystallite size accurately
4- Line # 36, “….Cu-based alloys, Ni-based superalloys, Fe-based steels, etc.”, references are missing
5- Line # 73, the abbreviation already explained in the abstract
6- Line # 78, the abbreviation already explained in the abstract
7- SPS already abbreviated line # 16, line # 73
8- The methodology seems interesting to me. I did not find any reference for the method from author. I would suggest the author to add a schematic diagram of the process for others to easily understand and follow!
9- Line # 134-135, please mention the operating voltage of SEM, is it 20 kV or what? also mention the EHT voltage for the measurement of sample
10- From XRD results, it will also be interesting for readers to know the origin of fcc structure and then conversion of bcc into mixed phases such as bcc and fcc as proposed by authors.
11- From XRD data in Fig. 2 and 3, the peak intensities (110, 111, 200, 220 & 211) in the case of SPS at 1073 K have significantly increased. Author did not explain the reason for the increment
12- Line # 249 abbreviation, M(T) should be explained here
13- Figure 6(a), I believe an MT curve slips into FC and ZFC, however, I did not see any such behavior since the material is showing a transition from fero- to paramagnetic, any thoughts?
14- Line # 282-283, it will be helpful to the readers, if authors can please add the formulation to calculate the magnetic moment per atom
15- Resolution of Fig. 6,7,8 and 9 must be improved

Author Response
We thank the Reviewer for the valuable and helpful comments and overall positive evaluation of the manuscript - with a recommendation for acceptance. We revised the manuscript according to the suggestions of the reviewers. Below, you will find our point-by-point response. The respective changes in the text are marked in yellow We hope that the revised version of the manuscript can be accepted for publication without further delay.

Round 2
Reviewer 1 Report
The authors answered all questions, the article can be accepted for publication.